# Lipophagy and Lipolysis Status in Lipid Storage and Lipid Metabolism Diseases

**DOI:** 10.3390/ijms21176113

**Published:** 2020-08-25

**Authors:** Anna Kloska, Magdalena Węsierska, Marcelina Malinowska, Magdalena Gabig-Cimińska, Joanna Jakóbkiewicz-Banecka

**Affiliations:** 1Department of Medical Biology and Genetics, Faculty of Biology, University of Gdańsk, Wita Stwosza 59, 80-308 Gdańsk, Poland; anna.kloska@ug.edu.pl (A.K.); magdalena.wesierska@phdstud.ug.edu.pl (M.W.); marcelina.malinowska@ug.edu.pl (M.M.); 2Laboratory of Molecular Biology, Institute of Biochemistry and Biophysics, Polish Academy of Sciences, Kładki 24, 80-822 Gdańsk, Poland

**Keywords:** lipophagy, lipolysis, lipid metabolism, lipid droplets, lipid storage diseases, lipid metabolism diseases, mTORC1, TFEB

## Abstract

This review discusses how lipophagy and cytosolic lipolysis degrade cellular lipids, as well as how these pathway ys communicate, how they affect lipid metabolism and energy homeostasis in cells and how their dysfunction affects the pathogenesis of lipid storage and lipid metabolism diseases. Answers to these questions will likely uncover novel strategies for the treatment of aforementioned human diseases, but, above all, will avoid destructive effects of high concentrations of lipids—referred to as lipotoxicity—resulting in cellular dysfunction and cell death.

## 1. Introduction

Lipids are water-insoluble biological macromolecules that are essential for maintaining cellular structure, function, signaling and energy storage. They are basic components of all cellular membranes which separate cell compartments in eukaryotic cells and provide a permeability barrier. These membrane boundaries are necessary for maintaining cellular homeostasis [1,2]. Moreover, lipids affect the function of membrane proteins. Lipid rafts play a specific role in protein segregation; membrane proteins can interact with lipids, which serve as cofactors [3,4]. Finally, changes in lipid organization influence signal transduction and membrane trafficking [2]. Cholesterol serves as a precursor for steroid hormones and bile acid biosynthesis [5]. Lipids are also molecules that serve as a source of energy when tissue energy is depleted [6]. Despite their role in essential cellular functions, incorrect lipid distribution or metabolism can result in abnormal concentrations of lipids being toxic because of their limited solubility and amphipathic nature, their adverse impact on cellular homeostasis and their ready transformation into highly bioactive, cytotoxic lipid species. These effects have serious consequences for cellular function and homeostasis and may even lead to cell death [2].

In this review, we provide information about lipid metabolism in health and disease, focusing on lipid storage diseases and lipid metabolism diseases. We summarize the current knowledge about the role of two cytosolic pathways designed for lipid selective catabolism—lipophagy and lipolysis—and discuss the transcriptional regulation of these processes by the mechanistic target of rapamycin kinase complex 1 (mTORC1)—transcription factor EB (TFEB) signaling. We also characterize lipid storage and lipid metabolism diseases, highlighting the latest research on the contribution of mTORC1-TFEB signaling in the regulation of lipophagy, a subtype of macroautophagy, and lipolysis, an enzymatic hydrolysis process, in the selected human dysfunctions.

## 2. Lipids in Eukaryotic Cells

Based on the chemical origin (i.e., whether ketoacyl groups or isoprene groups serve as fundamental “building blocks”), lipids are divided into eight categories: fatty acids, glycerolipids, glycerophospholipids, sphingolipids, saccharolipids, polyketides, sterols and prenols [7].

### 2.1. Fatty Acids and Cholesterol—Essential and Toxic

Fatty acids (FAs) are hydrophobic molecules consisting of an aliphatic hydrocarbon chain terminating in a carboxylic acid moiety. FAs usually contain 16–18 carbons, and the chain can be fully saturated (saturated FA) or may contain one or more double bonds (unsaturated FAs). The main source of FAs for humans and other animals are dietary fats and oils, but they can also be synthesized de novo from metabolites of sugar and protein catabolic pathways [8]. Fatty acids can be harmful to cells because of lipotoxicity; thus, cells convert FAs into neutral lipids for storage in organelles called lipid droplets (LDs). Biogenesis of LDs is stimulated upon the increase in cellular free FA levels. Different cell types have LDs of various sizes and numbers, potentially reflecting the capacity of the cell for managing lipid storage. Moreover, these organelles are often heterogeneous within a population of a single cell type. It is believed that LDs not only serve as lipid storage organelles, but also interact with most, if not all, cellular organelles, mediating lipid transfer via direct contact [9].

Cholesterol is an elementary component of mammalian cell membranes. It interacts with phospholipids and sphingolipid fatty acyl chains in order to maintain appropriate membrane fluidity. Interactions between these lipids also regulate water and ion membrane permeability [10]. Cholesterol is required for normal prenatal development, as embryonic and fetal cells demonstrate high membrane formation rates [11,12]. At both fetal and adult stages of development, cholesterol is the precursor for biosynthesis of five major classes of steroid hormones (i.e., androgens, estrogens, glucocorticoids, mineralocorticoids and gestagens), vitamin D and bile acids [10,11,13]. Mammalian cells require cholesterol for proliferation. Moreover, cholesterol is specifically required for the transition from G1 to S during cell cycle progression [14,15]. Additionally, cholesterol is essential for mitosis progression and its deficiency leading to aberrant mitosis and polyploid cell formation [15,16].

The cell synthesizes cholesterol de novo or internalizes it from exogenous sources. Interestingly, cells do not have any enzymes to break down the sterol core to acetyl-CoA units; thus, cholesterol cannot be used as an energy source [10]. Regardless of its source, free cholesterol must be esterified; otherwise, it has a toxic effect on cellular membranes and induces cell death. Esterified cholesterol is stored in cells in cytoplasmic lipid droplets [2,5].

### 2.2. Lipid Droplets—Storage of Neutral Lipids

LDs are highly dynamic cellular organelles responsible for the storage of neutral lipids. They are found in most eukaryotic cell types. The size of LDs varies within the range 0.4–100 µm in different cell types or within the same cells, depending on physiological conditions. Lipid droplets originate from the endoplasmic reticulum (ER) and have a unique architecture consisting of a hydrophobic core of neutral lipids which is enclosed by a phospholipid monolayer with hundreds of resident and transient proteins that influence LD metabolism and signaling, known generically as perilipins (PLINs) (Figure 1A). Organization of these organelles is quite different than any other because the core of a LD is hydrophobic, the hydrophobic acyl chains of the monolayer’s phospholipids are in contact with neutral lipids and the polar head groups face the aqueous cytosol [17]. Furthermore, LDs can also be found in the nuclei, where they are thought to regulate nuclear lipid homeostasis and modulate signaling through lipid molecules [18]. Cells preserve lipids by converting them into neutral lipids such as triacylglycerols (TAGs) and sterol esters (ESs), which are in various ratio deposit in LDs. Depending on the cell type, many other endogenous neutral lipids, such as retinyl esters, ether lipids and free cholesterol, can be stored in the LD core. Defects in LD biogenesis lead to insufficient or excess storage. Beyond the main function in energy metabolism, LDs play an important role in various cellular events, such as protein degradation, sequestration of transcription factors and chromatin components, generation of lipid ligands for certain nuclear receptors and serving as fatty acid trafficking nodes [19].

In mammalian cells, the phospholipid composition of LD membranes differs from that of the ER and other organelles. The main constituent is phosphatidylcholine (PC), followed by phosphatidylethanolamine (PE), phosphatidylinositol (PI), phosphatidylserine (PS) and sphingomyelin (SM), as well as free cholesterol and phosphatidic acid in minor amounts. The unique phospholipid membrane composition affects LD synthesis, size and catabolism. The homeostasis of these organelles under physiological conditions is maintained through changes in membrane phospholipid ratios in various cell types [20].

In addition to the composition of phospholipids, LD membrane surface proteins are another key factor that is important for their homeostasis and intracellular interactions. Each LD has many different structural and functional proteins on its surface. In mammalian LDs, predominant proteins are PLINs, adipophilin (ADRP) and a tail-interacting protein of 47 kDa (TIP47)—all belonging to the PAT (PLIN/ADRP/TIP47) protein family, which was named after its members [21]. Of these proteins, the structure and function of PLINs that regulate lipase access to the LD core is best known, and increased lipolysis in adipocytes is observed in their absence [22]. Furthermore, there are many other proteins involved in the maintenance of lipid homeostasis, including signaling and membrane trafficking proteins, chaperones and proteins associated with cellular organelles. Mitogen-activated protein kinase (MAPK) and phosphatidylinositol 3-kinase (PI3K) play the major role among signaling proteins associated with the LD surface [23]. Caveolin 1 (CAV1) and 2 (CAV2) are other proteins present on the LD surface; they generate membrane domains that serve as regulators of signaling proteins. In general, caveolins form a coat by making invaginations in surrounding cellular membranes. The coats are called caveolae and they function in endocytosis, signal transduction, cholesterol transport and growth control [24]. Amongst membrane trafficking-related proteins, five subgroups are distinguished: small GTPases governing vesicle formation and motility; proteins that carry LDs on the cytoskeleton, such as kinesin and myosin; proteins that mediate membrane docking and fusion, such as soluble N-ethylmaleimide-sensitive factor attachment receptor (SNARE); proteins that regulate cargo sorting and vesicle budding, such as ADP-ribosylation factor (ARF)-related proteins and coat proteins (COPs); and other proteins of miscellaneous function [19].

Generally, once LDs are synthesized they keep growing because of excessive amounts of intracellular FAs until they reach a final size. It has been shown that many proteins, e.g., PLIN1 and lipids, such as PC, are involved in LD growth mechanisms [25].

## 3. Lipophagy and Lipolysis—Two Pathways that Play a Crucial Role in Lipid Metabolism

Mobilization of fat stores from LDs is regulated by the metabolic and energy demands of the cell. This process usually appears in the form of lipophagy or lipolysis. They are the catabolic pathways that have a fundamental role in breaking down lipids during nutrient deprivation. Both have an impact on cellular energetic balance: directly through their important role in the early steps of lipid breakdown and indirectly by regulating food intake. Defects in lipophagy and lipolysis have been linked to many metabolic disorders; among them are lipid storage and lipid metabolism diseases.

### 3.1. Catabolism of Lipid Droplets

Catabolism of LDs into free FAs is a crucial cellular pathway that is required to generate energy in the form of ATP. Their catabolism is strictly under the control of hormone and enzyme activation. Moreover, it is required to provide building blocks for biological membranes and precursors in hormone synthesis. Degradation of LDs is strictly regulated by the protein composition on the surface of the vesicle and generally occurs by two mechanisms: lipolysis or lipophagy.

Lipolysis is a biochemical catabolic pathway that relies on direct activation of LD-associated lipases, such as adipose triglyceride lipase (ATGL), hormone-sensitive lipase (HSL) and monoglyceride lipase (MGL), which, together with regulatory protein factors (ATGL activators and inhibitors), constitute the basis for this process [26]. Under fed conditions, LDs mainly store TAGs in adipose tissue, and lipolytic hydrolysis is based on the hydrolysis of ester bonds between long-chain FAs and the glycerol backbone. During the first step of this process, protein kinase A (PKA) phosphorylates PLIN1, leading to its proteasomal degradation. This results in the release of an ATGL activator protein—comparative gene identification-58 (CGI-58)—which selectively activates ATGL, which then catalyzes TAG hydrolysis to diacylglycerols (DAGs) and free FAs. The next step of the process depends on the activation of a multifunctional enzyme, hormone sensitive lipase (HSL), that hydrolyzes DAGs and produces monoacylglycerol (MAG) and FAs (Figure 1B). HSL functions as a rate-limiting enzyme for DAG catabolism. HSL also retains specificity to other lipid ester bonds, such as cholesteryl esters, retinyl esters and short-chain carbonic acid esters [27]. It is responsible for mediating the hydrolysis of diacylglycerol and triacylglycerol. In testis, HSL is the only esterase that can hydrolyze cholesteryl ester, and the loss of this activity results in cholesteryl ester and diacylglycerol accumulation [28], as well as altered lipid homeostasis [29]. In the last step of the lipolysis cascade, MAGs are released into the cytosol and cleaved by MGL, generating glycerol and FAs [30,31]. Products of lipolysis secreted from adipose tissue are transported to other tissues and used for β-oxidation and ATP production. In non-adipose tissues, mitochondria or peroxisomes can directly oxidize products of lipolytic hydrolysis through β-oxidation and release acetyl CoA [32].

In turn, the lysosomal–autophagic pathway that plays an important role in the early steps of lipid degradation has been termed lipophagy. In general, autophagy is one of the major degradation pathways that enables the cell to survive under stress conditions by recycling metabolic components. This process is initiated by sequestering cytosolic organelles or macromolecules in a double membrane vesicle, which is then delivered to lysosomes for degradation by lytic enzymes. Degradation products that can be reused by the cell in synthesis processes are then released into the cytosol. Proper functioning of autophagy allows the cell to maintain homeostasis [33]. Due to the mechanism of the process, we can distinguish three types of autophagy: macroautophagy, which targets large substrates in a selective or nonselective manner to form autophagosomes prior to fusion with lysosomes [34]; microautophagy, which degrades molecules through direct engulfment by membranes of lytic compartments (lysosomes or late endosomes); and chaperone-mediated autophagy (CMA), which is a selective form of autophagy, targeting specific proteins through the recognition activity of chaperone protein heat shock cognate 70 (Hsc70) [35].

Uptake of LDs by macroautophagy is an alternative route for the mobilization of lipid storage and degradation of intracellular LDs (Figure 1C). Such a process is called lipophagy, in which LDs are selectively delivered to a lytic compartment for degradation via actions of autophagic (Atg) proteins. This process was first described in mouse hepatocytes under starvation, when LDs were mobilized in order to generate free FAs [36]. LC3, a classical marker of the autophagosome, was able to directly interact with ATGL and HSL at the surface of LDs. LC3 binds ATGL via an LC3 interaction region (LIR) and, under fed deprivation and LIR deficiency conditions, reduced basal ATGL localization to LDs, preventing the ATGL translocation to the LD surface, was observed. When we consider the above, it seems that LC3 is required for translocation of ATGL to the surface of LDs, to facilitate TAG hydrolysis [37].

Numerous Rab proteins have been identified on LDs. In general, Rab proteins are a family of small GTPases, acting as important mediators of endosomal trafficking events. They are molecular switches, cycling between active GTP-bound and inactive GDP-bound states, regulating the vesicular trafficking network within the cell. Perturbation to some members of the Rab family proteins has deleterious effects on LD turnover in response to classical lipophagy-inducing causes [38]. The most predominant Rab protein on the LD surface is Rab7, which is a well-characterized marker of the late endocytic pathway and a participant in the process of autophagosomal maturation. This protein assists in the regulation of lysosome–autophagosome interaction. Rab7 GTPase on the surface of LDs becomes active upon nutrient deprivation, resulting in its increased activity for GTP over GDP. Such an activated state promotes the requirement of lysosomes near LDs and their target degradation via lipophagy [39]. Another LD-localized protein from the Rab family that potentially participates in lipophagy is Rab10, which in its active state is significantly redistributed on the LD surface under nutrient deprivation conditions. This GTPase co-localizes with autophagic membrane markers such as LC3 and Atg16. However, it seems that Rab10 acts downstream of Rab7 as a part of a complex that promotes the envelopment of LD during lipophagy progression [40]. There are several other Rab proteins that have been studied to determine their role in LD catabolism via conventional lipolysis and selective lipophagy, such as Rab32, Rab18 and Rab25.

In LD catabolism, a link between PLIN proteins and CMA has been identified (Figure 1D). For the CMA process, LAMP2A is required and lack of LAMP2A leads to LD accumulation. Moreover, Hsc70 binds to CMA recognition motif (KFERQ) within PLIN proteins, acting as a signal for CMA-mediated degradation in the lysosome. It appears that degradation of PLIN proteins is required to promote LD catabolism by allowing ATGL and autophagic proteins to access the LD surface. Blocking the CMA process reduces lipase-mediated lipolysis and lipophagy. Therefore, CMA-mediated degradation of PLIN proteins seems to be a crucial event for initiating lipophagy [41,42].

### 3.2. Energy Release from Fatty Acids

Triacylglycerols are highly concentrated forms of metabolic energy because they are reduced and anhydrous. They are made up of three FAs that are ester-linked to a single glycerol. Complete oxidation of FAs provides more than twice as much energy than is obtained from carbohydrates or proteins. TAGs have much lower toxicity compared to FAs; thus, they can reach much higher concentrations (e.g., in plasma). For this reason, TAGs are the major form of FA storage and transport [6,43]. 

Before FAs can be used as a source of energy, they must be released from TAGs by lipolysis or lipophagy. Products of lipid stores that are broken down by lipolysis or lipophagy are subsequently utilized in β-oxidation for ATP production (Figure 2). At the outer mitochondrial membrane, FAs are activated by thioesterification to acyl-CoA esters. Next, carnitine, together with acylcarnitine translocase, transports FAs across the inner mitochondrial membrane to the matrix, where β-oxidation takes place. FA β-oxidation consists of four cyclic biochemical reactions. First, acyl-CoA is oxidized with the participation of flavin-adenine dinucleotide (FAD). At this stage, electrons from FADH_2_ reduce ubiquinone to ubiquinol, which transfers them on the respiratory chain and leads to the generation of 1.5 ATP molecules. Acyl-CoA oxidation introduces a double bond in the FA chain, which is hydrated in the second step of β-oxidation. The third reaction involves oxidation with creating ketone group at C-3 and reduction of NAD^+^ to NADH (to generate 2.5 ATP molecules). In the fourth and final step, the thiol group of the next CoA molecule resolves 3-ketoacyl-CoA into acyl-CoA (two carbon atoms shortened) and acetyl-CoA by thiolytic cleavage. Β-oxidation is repeated until the initial FA chain is converted into acetyl-CoA, which enters the Krebs cycle (to generate 10 ATP molecules) [44,45].

Mitochondrial β-oxidation is more complex with unsaturated FAs or odd-chain FAs being the source of energy. Degradation of unsaturated FAs involves the participation of additional enzymes. Isomerase converts configuration of the double bond in mono- and poly-unsaturated FA. Next, mono-unsaturated FA is hydrated, and β-oxidation progresses as for saturated FA. In turn, poly-unsaturated FA is oxidized with the participation of FAD and then reduced by mitochondrial NADPH-dependent reductase. Another double bond is formed and FA is again converted by isomerase until a regular intermediate of β-oxidation pathway is obtained, which is entered into the cycle at the stage of hydration [46]. At the end of odd-chain FA β-oxidation, acetyl-CoA and propionyl-CoA are produced in place of two molecules of acetyl-CoA. Propionyl-CoA enters into the Krebs cycle after it has been converted into succinyl-CoA [47].

### 3.3. Transcriptional Regulation of Lipophagy, Lipolysis and Lipid Metabolism

Lipophagy control depends on several transcription factors, activators and nuclear receptors, which, in response to nutrient status, enhance or decrease the process of lipid breakdown, in order to support current energy demands of the cell. Expression of many autophagy- and lipophagy-related genes is controlled by transcription factors belonging to the microphthalmia (MiT/TFE) family. One of these factors is TFEB, which regulates not only the general autophagic process, but lipophagy and lipid metabolism, as well [48]. Phosphorylation and dephosphorylation of TFEB determines its cellular localization and activity. Both events are mainly controlled by the nutrient or lysosomal storage status of the cell [48]. Under nutrition-rich conditions, the phosphorylated form of TFEB is retained in the cytoplasm (Figure 3A). However, nutrient depletion or aberrant lysosomal storage results in dephosphorylation of TFEB, causing its translocation from the cytoplasm to the nucleus, where it induces the transcription of target genes (Figure 3B). Promoters of many lysosome-related genes share a common 10-base E-box-like palindromic sequence; they compose the coordinated lysosomal expression and regulation (CLEAR) gene network that TFEB directly targets and controls the transcription process of [49]. Chromatin immunoprecipitation assays identified over 600 endocytic genes regulated by TFEB; among them are genes related to lysosomal biogenesis and autophagy, as well as lipid catabolism. TFEB phosphorylation is mainly exerted by two kinases: mTORC1 (one of two complexes having mTOR kinase as a core component) or extracellular signal-regulated kinase 2 (ERK2, also known as MAPK1). MTORC1 is the main negative regulator of autophagy, acting in response to amino acids, growth factors or cellular energy status [50]. Under sufficient intracellular availability of nutrients, mTORC1 is activated and inhibits autophagy, but as nutrients are deprived, this kinase is switched off, leading to autophagy activation and inhibition of anabolic processes. Active mTORC1 phosphorylates TFEB, preventing its translocation to the nucleus and thus indirectly inhibiting autophagy as the TFEB-dependent transcription of genes related to autophagy and lysosomal biogenesis stays suppressed [51].

The role of TFEB in lipophagy was firmly revealed by Settembre et al. [52], as they demonstrated that TFEB regulates lipid degradation in the mouse liver. They showed that, upon nutrient depletion, TFEB deficiency lead to accumulation of LDs and impairment of FA oxidation in the liver [52]. In animals receiving a high-fat diet, overexpression of TFEB prevented the development of obesity and improved the metabolic syndrome phenotype by reducing abnormal levels of circulating triglycerides, cholesterol, glucose and insulin [52]. High TFEB activity was also able to revert the metabolic syndrome when it was already present [52]. A functional autophagic pathway was required to observe TFEB-mediated lipid degradation, as overexpression of TFEB was not able to decrease the lipid droplet number, liver weight gain or lipid content in livers of mice with blocked autophagy [52]. Upon starvation, TFEB enhanced the expression of genes related to lipid metabolism and lipophagy [52]. In mice, TFEB-dependent transcriptional upregulation of monocarboxylic acid, FA, ketone catabolism and FA oxidation pathways was observed [52]. Several genes involved in lysosome organization and autophagy were also upregulated by TFEB, upon reduced nutrient availability; these include ATPase subunits, proteases, membrane proteins and fusion proteins [52]. Additionally, TFEB downregulated gene expression of lipid biosynthetic pathways; these included pathways of steroid, lipid, isoprenoid and FA biosynthesis [52]. TFEB was shown to regulate expression of genes involved in several steps of lipid catabolism, including genes related to FA transport across the plasma membrane (e.g., *Cd36* and *Fabps*), β-oxidation of free FAs in mitochondria (e.g., *Cpt1*, *Crat*, *Acadl*, *Acads* and *Hdad*) and peroxisomes (*Cyp4a* genes) [52]. Moreover, TFEB controls its own expression in an autoregulatory feedback loop [52]. CLEAR elements are present in the *TFEB* gene promoter; thus, as TFEB translocates to the nucleus, it is able to bind to its own promoter, enhancing transcription of itself in response to cellular status [52]. Interestingly, the role of TFEB in lipophagy appears to be evolutionarily conserved. In *Caenorhabditis elegans*, a gene encoding the worm’s TFEB orthologue, *HLH-30*, regulates the expression of fat catabolism enzymes and autophagy genes in response to nutrient availability. Moreover, loss-of-function mutations of this gene result in impairment of lipophagy [53]. Transcriptional control of genes involved in lipid metabolism by TFEB is exerted through the peroxisome proliferator-activated receptor gamma coactivator 1α (PPARGC1α; also known as PGC1α) and the downstream nuclear receptor peroxisome proliferator activated receptor 1α (PPAR1α) [52]. The promoter of the PGC1α-encoding gene was shown to have three CLEAR sites and upon starvation TFEB bound to two of them [52]. Thus, TFEB induces PGC1α expression upon starvation and regulates the expression of genes related to lipid metabolism by controlling the downstream PPARα activity [52,54]. Independently from TFEB, lipophagy control is also mediated by nuclear farnesoid X receptor (FXR) and transcriptional activator cAMP response element-binding protein (CREB) [55]. Under nutrient-deprived conditions, CREB promotes lipophagy by the upregulation of autophagy-related genes, including *Atg7*, *Ulk1* and *Tfeb*, but FXR, a fed-state sensing gene regulator, inhibits this response after feeding [55].

Lipolysis, another process involved in intracellular lipid breakdown, was shown to be transcriptionally modulated by forkhead homeobox type O (FOXO) and TFEB transcription factors. Nutrient restriction upregulates FOXO1, which then activates lipid catabolism by inducing lysosomal acid lipase (LAL) expression [56]. In this case, colocalization of LDs with lysosomes was observed. In response to nutrient restriction, FOXO1 and TFEB were shown to induce the expression of *LIPA*, a gene encoding LAL. In a mouse atherosclerosis model, lysosomal stress conditions induced by atherogenic lipids were shown to promote TFEB translocation to the nucleus and upregulation of *LIPA* gene expression and lysosomal biogenesis [57]. TFEB overexpression in macrophages loaded with acetylated low-density lipoprotein (LDL) enhanced cholesterol efflux [57].

Lipophagy and lipid metabolism are also regulated by transcription factor E3 (TFE3), which is another transcription factor belonging to the MiT family. In the liver, TFE3 was shown to induce lipophagy as its overexpression alleviated steatosis of this organ in mice [58]. Similarly to TFEB, TFE3 induces expression of genes that modulate mitochondrial fatty acid β-oxidation; an increased mRNA level of PGC1α and PPARα was found upon TFE3 overexpression [58]. Another study also showed that TFE3 deficiency resulted in altered mitochondrial morphology and function [59]. *Tfe3*-knockout mice show abnormalities in energy balance and alterations in systemic glucose and lipid metabolism, resulting in high-fat-diet-induced obesity and diabetes [59]. However, overexpression of TFE3, as well as TFEB, was shown to improve this metabolic outcome [59]. Because TFE3 and TFEB were able to compensate for each other’s deficiency, the authors suggested that both play a cooperative role in controlling metabolism.

Generally, autophagy was thought to contribute to lipid oxidation by increasing the supply of free FAs by lipophagy. However, a recent study demonstrated that autophagy regulates lipid metabolism by participating in regulation of PPARα activity through the degradation of nuclear receptor co-repressor 1 (NCoR1) [60]. Autophagic degradation of NCoR1 was shown to contribute to PPARα activation to effectively promote β-oxidation in response to physiological fasting [60]. Defects of liver autophagy were accompanied by accumulation of NCoR1 and suppression of PPARα activity leading to defective β-oxidation and ketogenesis [60].

## 4. Lipid Metabolism and Diseases

Lipids have been found necessary in tissues such as adipose tissue, intestine and liver for energy storage or lipid turnover, but they are accreted in skeletal muscles, macrophages, mammary glands and the adrenal cortex. Under energy-poor conditions, lipid accumulation allows organisms to survive, and stored lipids are then used to produce energy [61]. Abnormal lipid metabolism is associated with many diseases, including type 2 diabetes, obstructive sleep apnea, non-alcoholic fatty liver disease, coronary artery disease and cancer. A number of studies have been published to reveal the important role of lipid metabolism in energy homeostasis and metabolic diseases [62]. Inherited metabolic disorders are genetic conditions that cause metabolism problems. There are hundreds of different genetic metabolic disorders, and their symptoms, treatment and prognosis vary significantly. Genetic disorders associated with abnormal lipid turnover belong to two groups of inherited metabolic disorders: lipid storage diseases and lipid metabolism diseases [63].

The pathological accumulation of undigested biomaterials in the lysosome, including lipids, leads to the development of metabolic disorders collectively called lysosomal storage diseases (LSDs). LSDs are traditionally classified due to the nature of undigested materials in the lysosome; in this group, mucopolysaccharidosis, cystinosis, mannosidosis and lipid storage diseases can be distinguished.

### 4.1. Characterization of Lipid Storage Diseases and Lipid Metabolism Diseases

Lipid storage diseases are the most common LSDs and constitute the largest group of these diseases [64]. However, undigested lipids can also accumulate due to secondary mechanisms, e.g., mechanisms secondary to carbohydrate or protein accumulation or membrane trafficking. Most LSDs are caused by lysosomal hydrolase mutations. Lipid storage diseases are a genetically determined group of disorders characterized by excessive lipid (fatty acids, cholesterol or complex lipids) accumulation due to inherited abnormalities in lipid metabolism. Excessive lipid deposition ultimately causes damage to cells and tissues, resulting in neurodegeneration and also often heart, liver, spleen and kidney problems [65]. In most lysosomal lipid storage diseases, the accumulation of one or more lipids leads to the co-precipitation of other hydrophobic substances in the endolysosomal system, such as lipids and proteins [66]. The progressive accumulation of undigested lipids in lysosomes leads to the accumulation of enlarged (>500 nm) but dysfunctional lysosomes [67]. These swollen lysosomes are mainly endolysosomes and autolysosomes; therefore, LSD is a state of endocytic and autophagic “block” or “arrest”. Although the total number of lysosomes is not reduced in LSDs, the overall lysosomal function in the cell is blocked, which can lead to serious cell consequences [68]. The accumulation of undigested materials in lysosomes can cause a deficiency of building block precursors for biosynthetic pathways, while the accumulation of various membrane-associated lipids can affect the properties and integrity of the cell membrane. In addition, lipid storage may alter the functionality of lysosomal membrane proteins, such as lysosomal ion channels or catabolite exporters, affecting the physiology and ionic composition of lysosomes. In turn, altered heavy metal ion homeostasis can increase oxidative stress, causing lipid peroxidation and affecting membrane integrity. Over time, excessive lipid storage can cause permanent damage to cells, tissues in the brain and peripheral nervous system and other organs [64]. The brain is particularly sensitive to lipid accumulation, as any increase in fluid or deposits can lead to changes in pressure and disruption of normal neurological function [69]. Symptoms may appear early in life or develop in teenage years or even adulthood. Neurological complications of lipid storage diseases depend on the type of storage material and may include: lack of muscle coordination, brain degeneration, seizures, loss of muscle tone, spasticity, difficulty feeding and swallowing, pain in arms and legs and corneal opacity [70].

Congenital lipid metabolism errors are a heterogeneous group of disorders characterized by problems with the breakdown synthesis of lipids. Diseases that affect lipid metabolism can be caused by defects in the structural proteins of lipoprotein particles, in the cell receptors that recognize different types of lipoproteins or in fat-breaking enzymes [71]. As a result of such defects, excessively accumulating lipids can deposit in the walls of blood vessels, which can lead to atherosclerosis and, as a consequence, strokes or coronary heart diseases. Lipid metabolism diseases are associated with an increase in plasma lipoprotein levels, such as LDL cholesterol, very low-density lipoprotein (VLDL) and triglycerides, or combinations thereof. The historical framework for the classification of lipoprotein disorders is dominated by the Fredrickson classification, which is based on the pattern of lipoproteins on electrophoresis or ultracentrifugation [72]. The phenotypic classification of lipid metabolism diseases, which is widely used and has been accepted internationally, is based on the affected lipoprotein; however, the clinical approach is to classify dyslipidemia according to the high lipid content fraction: cholesterol (hypercholesterolemia, Fredrickson class IIa), triglycerides (hypertriglyceridemia, Frederickson classes I, IV and V) or a combination of the two (hypercholesterolemia and hypertriglyceridemia, Fredrickson classes III or IIb) [73]. In addition, it is crucial to consider the etiological aspects of dyslipidemias, which may help in the diagnosis or initial treatment.

Table 1 contains a list of examples of diseases classified as lipid storage diseases and lipid metabolism diseases. The classification was based on data contained in Mammalian Phenotype Ontology [63], with the exception of sitosterolemia. This disease is sometimes classified as rarer inherited metabolic disorder [72].

### 4.2. Dysregulation of Autophagy or Lipolysis in Diseases

Due to the important role of the lysosome in autophagy, this pathway is an obvious candidate in the pathogenesis of LSDs [167]. Autophagy has been identified as the primary pathway of lipid metabolism in cells [36]; therefore, it is believed that perturbances in autophagy, particularly lipophagy, are responsible for cellular lipid accumulation in patients with lipid storage diseases [168]. Studies have documented autophagic dysregulation in patient samples and disease models of various LSDs (i.e., Tay–Sachs disease, Fabry disease and Krabbe disease) [77,169]. Although impaired autophagy has been seen in many storage diseases, the defects observed relate to different stages of the autophagic pathway (Table 1 and Figure 4). While in GM1 gangliosidosis and Niemann–Pick disease, the impairment is due to overactivation of autophagy, in other LSDs, e.g., MSD and MLD, the autophagosome–lysosome fusion is defective [96]. Similar abnormalities of autophagy can be observed in diseases with secondary lipid accumulation (Figure 4). Although there is an increasing evidence of dysregulation of autophagy in lipid storage disorders, the role of these abnormalities in the pathogenesis of the diseases is still not well understood and requires further research. Impairment of autophagy has also been observed in lipid metabolism diseases; for example, autophagy induction occurs in the familial hypercholesterolemia and familial defective apoB-100. Impaired lipolysis has been reported in two groups of diseases—the neutral lipid storage diseases and the familial chylomicronemia syndromes (lipid metabolism diseases).

### 4.3. Secondary Lipid Accumulation in Lysosomal Storage Diseases

Storage processes in LSDs are much more complex than one would expect from the deficiency of a single enzyme altering a single substrate degradation in a particular catabolic pathway. Multiple substrates at variable ratios are detected as the storage material; this may result from metabolic links, e.g., when one enzyme is committed in the catabolism of multiple compounds. Secondary storage compounds can be actively involved in the pathogenesis of LSDs and the most common group of compounds that are subject to secondary storage are lipids. Phospholipids, glycosphingolipids and cholesterol are mainly identified as secondary storage materials. In numerous LSDs, one or more of these compounds, in various proportions, may accumulate inside cells (Table 2).

Two phospholipids, i.e., sphingomyelin and bis(monoacylglycero)phosphate (BMP), are identified as the secondary storage lipids in LSDs. Sphingomyelin is a primary storage material in Niemann–Pick type A and B, but cholesterol is the primary material in Niemann–Pick type C, while sphingomyelin is the secondary storage material. A moderate sphingomyelin increase already occurs in livers from 20-week-old fetuses with Niemann–Pick type C and remains at this level; sphingomyelin levels in the spleen are much more elevated, as compared to those in the liver [171]. Interestingly, the main organ of sphingomyelin accumulation in mice is the liver [182]. Currently, no secondary sphingomyelin storage has been identified in the brain from LSDs. Accumulation of BMP occurs in the liver and spleen of humans with all three types of Niemann–Pick disease, but it has not been identified in the brain [171,182]. BMP storage in the brain was described for humans with infantile neuronal-ceroid lipofuscinosis (CLN1 disease) [183] and for mouse models of other types of neuronal-ceroid lipofuscinosis, CLN6 and CLN10 diseases [184].

Secondary accumulation of GM2 and GM3 gangliosides is very often observed in diseases with progressive neurodegeneration. In the brains of healthy humans or wild-type mice, GM2 and GM3 constitute only 1–2% of total gangliosides in humans and even less in mice. In immunostaining, these gangliosides appear as punctate, granular structures in the cytoplasm, suggesting that they are sequestered in vesicles [171]. More precise analysis has shown that GM2 and GM3 are found in separate vesicle populations in the same cell; this may suggest that they are metabolized in separate cell compartments or are generated by independent processes [173]. GM2 and GM3 gangliosides accumulate in various neurons and glial cells in various LSDs, e.g., in Niemann–Pick type C [182]; mucopolysaccharidosis (MPS) types I, II, IIIA, IIIB, IIID, VI and VII [173]; and mucolipidosis type IV [185]. However, the presence of these gangliosides in non-nervous organs has been studied in only a few disorders. Elevated GM3 levels have been reported in the liver and spleen of patients with Niemann–Pick and Gaucher diseases [171].

Unesterified cholesterol is primary storage material in Niemann–Pick type C; this is in contrast to Niemann–Pick types A and B, where cholesterol is accumulated secondarily to sphingolipids. Sphingomyelin effectively inhibits the secretion of cholesterol from late endosomes and lysosomes, which results in secondary cholesterol storage. In turn, unesterified cholesterol affects sphingomyelin metabolism and regulates the trafficking of sphingolipids to other sites in the cell. Finally, perturbation in cholesterol homeostasis correlates with sphingolipid accumulation. Accumulation of unesterified cholesterol, which appears as storage-like granules inside cells, was also observed in MPS types I, II, IIIA and VI, as well as mucolipidosis types II and IV [171,185,186].

### 4.4. Consequences of Secondary Lipid Storage

Excessive accumulation of compounds in cellular compartments is often a pathological process. Storage of useless materials may be a result of a deficiency of catabolic enzymes, but disturbed catabolism may also occur without any apparent genetic defects leading to underlying enzyme deficiencies. When the primary storage in a particular disorder leads to secondary lipid accumulation, it may result in incorrect vesicular or protein trafficking, signal transduction and membrane disability.

Lysosomal storage of undegraded compounds leads to disturbed secretion of breakdown products from autolysosomes, consequently resulting in deficiency of precursors for cellular biosynthetic pathways. Lipid turnover has fundamental importance in maintaining membrane permeability to secure cell homeostasis. Cholesterol and glycosphingolipids are the main components of lipid rafts, which play an important role in determining membrane plasticity. Keeping the membrane plasticity is fundamental for fusion between autophagosomes and lysosomes in lipophagy. This process has an influence on lipid turnover that, as a consequence, may affect the properties and functionality of the membranes of other organelles. An example is a robust loss of mitochondrial membrane potential in multiple sulfatase deficiency (MSD) cells after starvation [110]. Abnormal lipid metabolism leads to disturbances in trafficking of synthesized proteins and lipids to their target destinations in the cell. Induction of cholesterol accumulation in Niemann–Pick type C cells perturbs the intra-endosomal trafficking [187]. Cholesterol stores correlate with primary or secondary storage of glycosphingolipids, suggesting that a molecular linkage between the sequestration of these two lipid classes may exist. In turn, blockade of GM2 and GM3 ganglioside synthesis results in an absence or dramatic reduction in free cholesterol in NPC1-deficient neurons [188]. Further investigations are needed to determine whether cholesterol sequestration depends on gangliosides or whether the storage of GM2 and GM3 gangliosides disturbs lipophagy.

Correct membrane function is also important for maintaining the physiology of the lysosome, the organelle crucial for lipophagy. Appropriate lysosomal pH and its regulation are essential for the activity of lysosomal acid hydrolases. Lipid dyshomeostasis may alter the functioning of lysosomal membrane proteins (e.g., V-ATPase), ion channels and catabolite exporters, affecting lysosomal physiology. Accumulation of primary storage molecules may inhibit catabolic pathways that are genetically unaffected, and, as a consequence, accumulation of that pathway’s substrates as secondary storage materials begins. For example, primary storage of chondroitin sulfate (in mucopolysaccharidoses—Hurler disease, Hunter disease, Sanfilippo disease and Sly syndrome) or sphingomyelin (in Niemann–Pick types A and B) and cholesterol (in Niemann–Pick type C) inhibits several catabolic pathways of gangliosides and glycosphingolipids and causes secondary neuronal GM2 accumulation that triggers neurodegeneration [189,190]. Furthermore, primary sphingomyelin storage in Niemann–Pick type A and B results in strong inhibition of lysosomal cholesterol secretion [191], which may lead to its deficiency in internal circulation. Excessive material storage inside the lysosome affects lysosomal pH and impairs activities of various acid hydrolases, consequently leading to the accumulation of other components. Dysfunctional lysosomes also undergo defective fusion with autophagosomes, which has been observed in cell culture models or macrophages of MPS IIIA and MSD [110].

### 4.5. mTOR–TFEB Signaling Pathway and Dysregulation of Autophagy in Lipid Storage Diseases

Functional lysosomes are particularly important for autophagy; lysosomes, by fusing with autophagosomes, deliver digestion enzymes that are necessary for breaking down the stores. Disruption of the autophagy–lysosomal pathway affects the normal autophagic flux and leads to impaired cellular capacity to remove the stored materials. Deregulation of autophagy has been reported in many lysosomal storage diseases, including those characterized with lipids as the main storage material—lipid storage diseases.

Dysfunction of the autophagy–lysosomal pathway is indicated as the main pathogenic event associated with neurodegeneration in Gaucher disease. Impaired autophagosome maturation accompanied with downregulation of TFEB and reduction of lysosomal gene expression were found in neurons differentiated from induced pluripotent stem cells of Gaucher disease patients [93]. MTORC1 was shown to be hyperactivated by the accumulation of glycosphingolipids in Gaucher cells [192]. As a consequence, increased TFEB phosphorylation by mTORC1 lead to decreased TFEB stability in Gaucher cells. The authors proposed that glycosphingolipid accumulation in Gaucher disease leads to increased mTORC1 activity, which in turn results in increased TFEB phosphorylation. Phosphorylated TFEB is targeted for proteasomal degradation and downregulation of lysosomal functions is observed as a consequence. It was already shown that mTORC1 regulates lipid metabolism [193] by controlling lipophagy in response to the nutrition status of the cell [123]; this process is mediated by mTORC1 and TFEB. It is therefore possible that the lipid storage disrupts the proper signaling of autophagy/lipophagy pathways.

In Fabry disease, a disturbance of the autophagic pathway is observed in kidney cells, fibroblasts and lymphoblasts [86,194]. Interestingly, studies on female Fabry disease cases showed that mild symptoms correlate with normal autophagic flux, whilst severe symptoms correlate with abnormal autophagic flux with enlarged lysosomes [195]. Neuropathology and axonal neurodegeneration in a Fabry disease mouse model was shown to be associated with disruption of the autophagy–lysosome pathway [196]. Accumulation of intracellular globotriaosylsphingosine was found to cause increased autophagosome formation, loss of mTOR kinase activity and downregulation of Akt kinase activity in Fabry podocytes, suggesting that dysregulated autophagy in Fabry disease may result from deficient mTOR signaling, which possibly leads to podocyte damage [194]. Other studies on both Fabry and Gaucher disease blood mononuclear cells revealed that dysfunction of the mTOR pathway accompanies sphingolipid accumulation, but was shown to be partially improved by enzyme replacement therapy [197]. 

In mice, Niemann–Pick type C1 maturation of autophagosomes appears to be impaired due to defective amphisome formation caused by the failure in soluble N-ethylmaleimide-sensitive factor attachment receptor (SNARE) machinery [81]. Decreased cell viability, cholesterol accumulation and dysfunctional autophagic flux was characteristic for Niemann–Pick-type-C1-deficient human hepatic and neural cells [198]. Genetic correction of a disease-causing mutation rescued these defects and directly linked NPC1 protein function to impaired cholesterol metabolism and autophagy [198]. Recently, cholesterol was identified as an essential activator for the master growth regulator, mTORC1 kinase. Cholesterol promotes mTORC1 recruitment and activation at the lysosomal membrane, and a lysosomal transmembrane protein called SLC38A9 is required for this process [199]. Lysosomal cholesterol content was shown to regulate mTORC1 signaling in Niemann–Pick type C. ER–lysosome contacts enable cholesterol sensing by mTORC1, as was shown in a Niemann–Pick type C1 model [200]. Cholesterol trafficking mediated by oxysterol binding protein (OSBP), a protein responsible for cholesterol delivery across ER–lysosome contacts, results in constitutive mTORC1 activation in a Niemann–Pick type C model, while cells lacking OSBP show inhibition of mTORC1 recruitment by Rag GTPases as a result of impaired transport of cholesterol to lysosomes [200].

Increased autophagy was demonstrated in GM1-gangliosidosis mouse brains, and it was accompanied with enhanced Akt-mTOR and ERK signaling [201]. In this case, activation of autophagy was pointed to lead to mitochondrial dysfunction in the mouse brain as the mitochondria isolated from animals were morphologically abnormal and had a decreased membrane potential.

In mucolipidosis type IV, mTOR kinase directly targets and inactivates the transient receptor potential mucolipin 1 (TRPML1) channel, a lysosomal calcium channel, mutations of which cause this disease, thereby affecting functional autophagy [202]. Lysosomal calcium release through TRPML1 channel was shown to regulate autophagy by promoting TFEB dephosphorylation by calcineurin [203]. TRPML1 channel was also shown to regulate autophagosome biogenesis by a mechanism independent of TFEB. TRPML2 can act through activation of a signaling pathway of calcium/calmodulin-dependent protein kinase β (CaMKKβ) and AMP-activated protein kinase (AMPK), the induction of the Beclin1/VPS34 autophagic complex and the generation of phosphatidylinositol 3-phosphate (PI3P) [204].

Neuronal ceroid lipofuscinoses are also characterized with inhibition of autophagosome formation, reduction in autophagosomes and autophagic degradation, defects in autophagosome maturation, accumulation of autophagosomes and autophagic cargo [117,118,205]. Mechanisms involved in the autophagy deregulation include upregulation of mTOR signaling [120], intracellular calcium homeostasis and CLN3 protein (also named battenin) function [206].

Other lysosomal lipid storage diseases are also associated with observations of impaired autophagy, but only a limited number of studies have been performed to elucidate the mechanism. For example, autophagy was shown to be defective in Tay–Sachs disease due to either a reduction in the number of autophagosomes produced or the amount of autophagic flux; studies on pyrimethamine, a known pharmacological chaperone of β-hexosaminidase A, showed that the mechanism of action of pyrimethamine in reversing the defective lysosomal phenotype was by improving autophagy [207]. Autophagy dysregulation is also observed in Krabbe disease; expression of some fundamental autophagy markers (LC3, p62 and Beclin-1) was elevated in the brain and sciatic nerve of a murine model of the disease [90]. Treatment with rapamycin, an autophagy inducer, was shown to restore autophagy in vitro [90].

## 5. Conclusions

To sum up, despite the major progress in our understanding of how the different pathways—lipophagy and lipolysis—communicate with each other, how they contribute—separately and collectively—to cytosolic degradation of lipids, how they affect the human pathophysiology and pathogenesis of lipid storage and lipid metabolism diseases, many questions remain unanswered. Over the past few years, an increasing body of research in this subject—as we reported in this review—has radically refilled our knowledge. However, the structural and functional depiction of the lipophagic and lipolytic machinery is still incomplete. Thus, the functional link between lipophagy and lipolysis and their cross-talk in the regulation of lipid metabolism to prevent and treat lipid accumulation and lipotoxicity requires further interrogation.

## Figures and Tables

**Figure 1 ijms-21-06113-f001:**
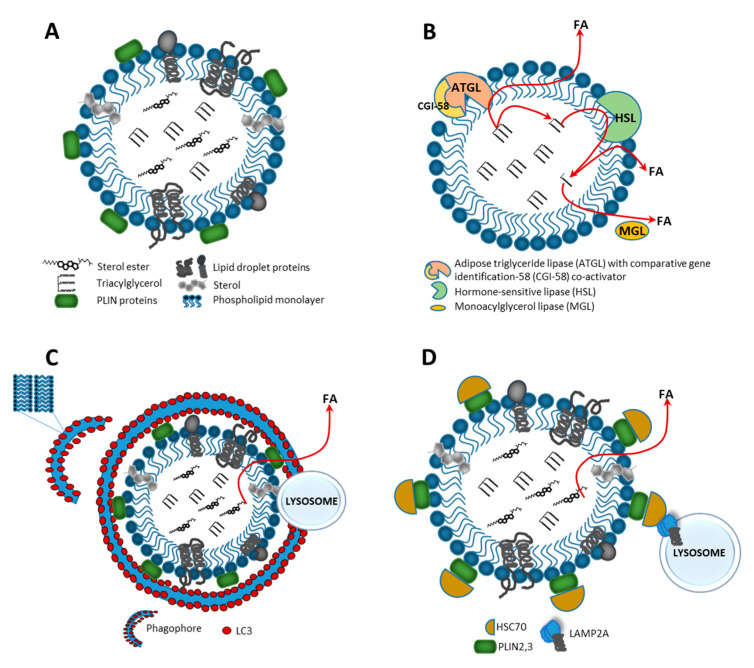
Structure and catabolism of a lipid droplet (LD). (**A**) LD is surrounded by the phospholipid monolayer enclosing a core filled with neutral lipids, e.g., triacylglycerol (TAG) and sterol esters. Polar heads of phospholipids are oriented toward the cytosol, whereas their acyl chains contact the hydrophobic lipid core. The LD surface is associated with various proteins, e.g., members of the perilipin (PLIN) family. There are two major types of LD catabolism: lipolysis—an enzymatic hydrolysis of lipids in cytosol, and lipophagy—an autophagic/lysosomal pathway in the form of macroautophagy or chaperone-mediated autophagy (CMA). (**B**) In lipolysis, protein kinase A (PKA) phosphorylates PLIN1 proteins, leading to their proteasomal degradation and activating adipose triglyceride lipase (ATGL), which then initiates TAG hydrolysis to generate diacylglycerols (DAGs) and free fatty acids (FAs). Further degradation of DAGs occurs through activation of the hormone sensitive lipase (HSL), leading to monoacylglycerol (MAG) and FAs production. MAGs are released to the cytosol and cleaved by monoacylglycerol lipase (MGL) to generate glycerol and FAs. (**C**) In macroautophagy, the phagophore is formed and LC3 positive membranes engulf small LD or sequester portions of a large LD to form the autophagosome, which later fuses with lysosome where LD degradation and neutral lipid catabolism occur. (**D**) In chaperone-mediated autophagy, lipid droplet-coat proteins—PLIN2 and PLIN3—are degraded through a coordinated action of Hsc70 protein and lysosome-associated membrane protein 2A (LAMP2A) receptor; this makes the LD surface accessible to cytosolic lipases, which hydrolyze LD cargo to generate FAs, which next are released to the cytosol and undergo subsequent mitochondrial β-oxidation.

**Figure 2 ijms-21-06113-f002:**
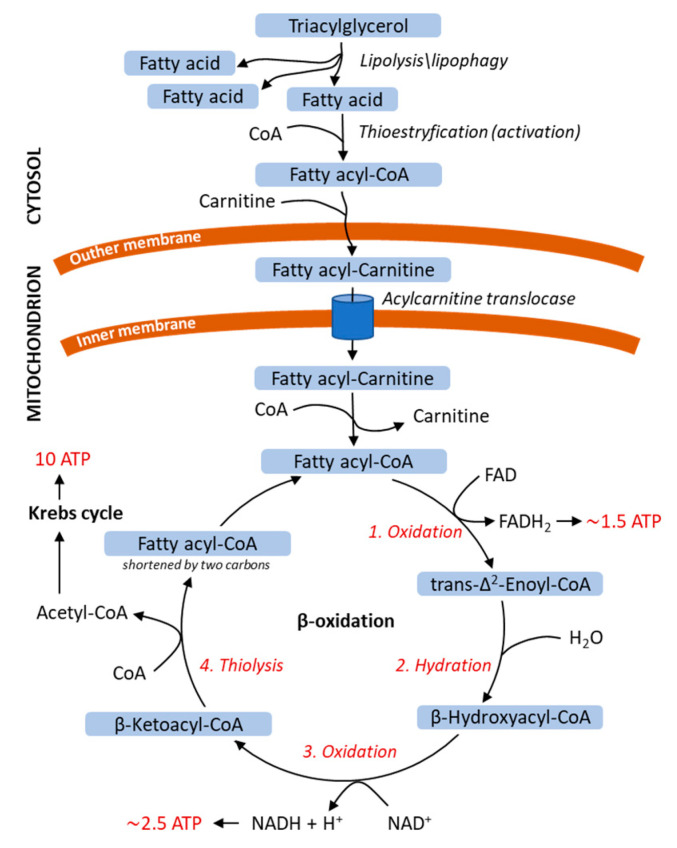
Energy release from saturated fatty acids in mitochondrial β-oxidation. Fatty acids are released from triacylglycerol by lipolysis or lipophagy and translocated into the mitochondrion. Fatty acid is shortened by two carbons in one β-oxidation cycle; the β-oxidation steps are repeated until only two carbon units remain. The FADH_2_ and NADH are utilized to generate ATP in the electron transport chain and acetyl-CoA enters the Krebs cycle. The β-oxidation steps are shown in red italics, numbered 1–4. The number of ATP molecules obtained from β-oxidation and Krebs cycle is shown in red. ATP, adenosine triphosphate; CoA, coenzyme A; FAD and FADH_2_, flavin-adenine dinucleotide, oxidized and reduced forms, respectively; NAD^+^ and NADH, nicotinamide adenine dinucleotide, oxidized and reduced forms, respectively.

**Figure 3 ijms-21-06113-f003:**
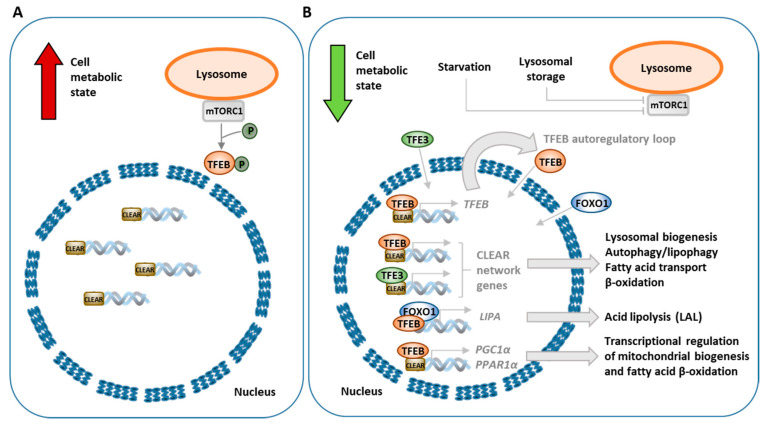
Transcriptional regulation of autophagy/lipophagy, lipolysis and lipid metabolism by transcription factor EB (TFEB) under nutrition-rich conditions (**A**) and nutrient depletion or aberrant lysosomal storage (**B**). Bold red arrow indicates the fed metabolic state during nutrient sufficiency, while bold green arrow shows low metabolic state due to nutrient limitations or abnormal lysosomal storage.

**Figure 4 ijms-21-06113-f004:**
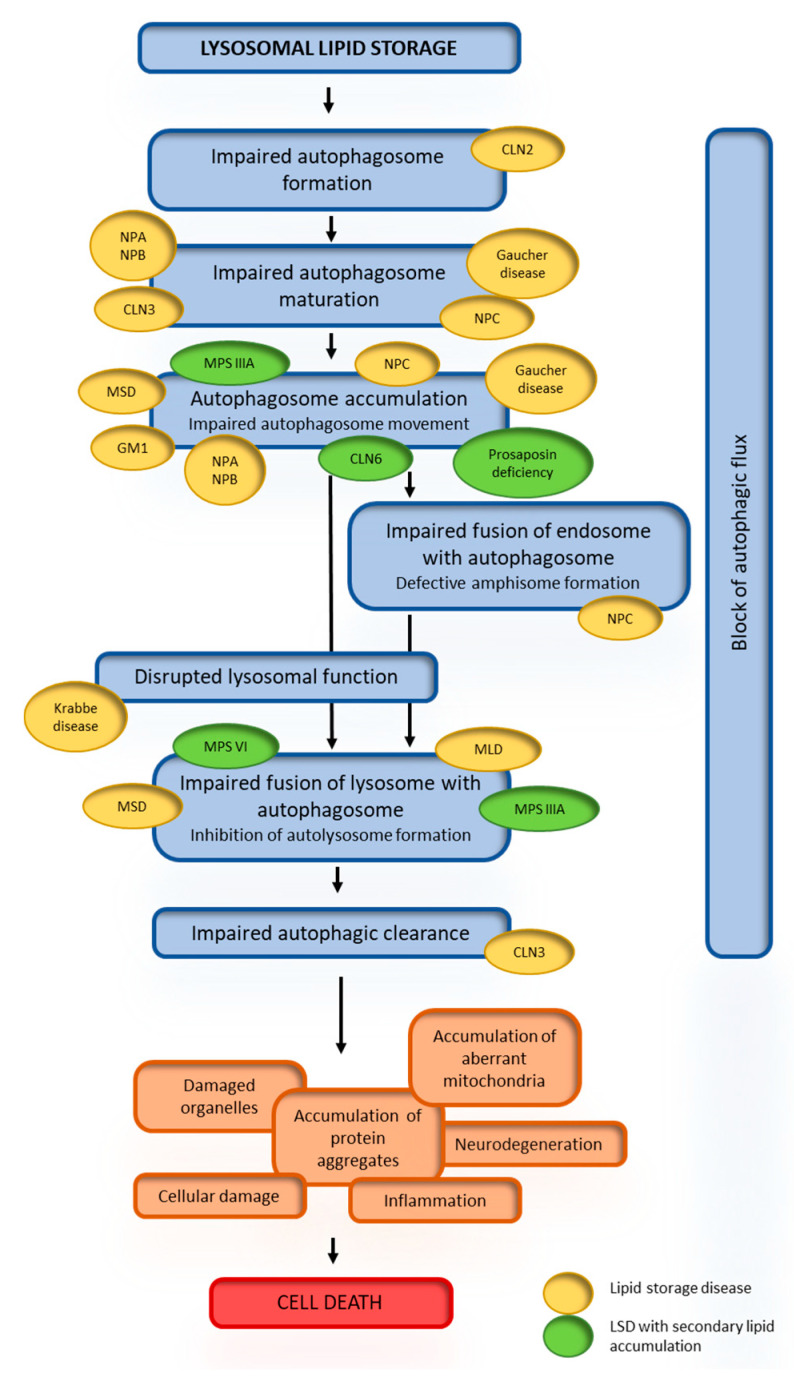
Alterations in different stages of autophagy in the pathogenesis of lipid storage diseases. Lysosomal lipid storage leads to a reduced ability to autophagosome formation, maturation, or fusion of lysosomes with autophagosomes. This results in a block of the autophagic flux. The steps of these abnormalities are presented in blue boxes. Consequently autophagy substrates (orange boxes) such as protein aggregates and dysfunctional mitochondria accumulate and promote cell death. The inflammatory response, cellular damage or neurodegeneration (orange boxes) further contribute to cell death (red box).

**Table 1 ijms-21-06113-t001:** Characterization of lipid storage diseases and lipid metabolism diseases.

Disease	Gene Deficient Enzyme/Protein	Accumulated Products	Symptoms	Perturbations in Autophagy/Lipophagy/Lipolysis	Reference
**Lysosomal storage diseases**					
Lipid storage diseases					
*Sphingolipidoses*					
Niemann–Pick disease types A and B	*SMPD1* sphingomyelinase	Sphingomyelin in brain and red blood cells (RBCs)	Hepatosplenomegaly, psychomotor regression, clumsiness and difficulty walking, dystonia, sleep disturbances, difficulty swallowing and eating, recurrent pneumonia, thrombocytopenia, a cherry-red spot inside the eye, frequent respiratory infections, slow mineralization of bone	Impaired autolysosomal clearance; formation of late endosome/lysosome (LE/LY)-like storage organelles (LSOs) and the misdirection of lipids to the LSOs; defect in autophagosome maturation; accumulation of autophagosomes	[74,75,76,77]
Niemann–Pick disease type C	*NPC1* or *NPC2* intracellular cholesterol transporters located within lysosomal and endosomal membranes (NPC1) or inside lysosomes (NPC2)	Free cholesterol, sphingomyelin and glycosphingolipid storage in lysosomes or late endosomes	Hepatosplenomegaly, problems with speech and swallowing, dementia, seizures, ataxia, vertical supranuclear gaze palsy, dystonia, severe liver disease, interstitial lung disease	Defective amphisome formation; impaired maturation of autophagosomes; accumulation of autophagosomes and autolysosomes	[78,79,80,81,82]
Fabry disease	*GLA* α-galactosidase A	Glycolipids, particularly ceramide trihexoside, in brain, heart and kidney	Episodes of pain (particularly acroparesthesias), angiokeratomas, hypohidrosis, corneal opacity or corneal verticillate, problems with the gastrointestinal system, tinnitus, hearing loss, kidney damage, heart attack, stroke	Impairment of the autophagic pathway	[83,84,85,86]
Krabbe disease (globoid cell leukodystrophy)	*GALC* galactocerebrosidase	Glycolipids, particularly galactocerebroside, in oligodendrocytes	Irritability, muscle weakness, feeding difficulties, stiff posture, delayed mental and physical development, spasticity, hypertonia, blindness, hyperreflexia, deafness, neurodegeneration (leading to death)	Impairment of autophagy; lysosomal dysfunction; partial blocking and saturation of the autophagy flux	[87,88,89,90]
Gaucher disease	*GBA* glucocerebrosidase	Glucocerebrosides in RBCs, liver and spleen	Hepatosplenomegaly, pancytopenia, Erlenmeyer flask deformity, anemia, lung disease, bone abnormalities such as bone pain, fractures, arthritis	Impaired autophagosome maturation; accumulation of autophagosomes; autophagy block	[91,92,93]
Tay–Sachs disease	*HEXA* β-hexosaminidase A	GM2 gangliosides in neurons	Neurodegeneration, seizures, vision and hearing loss, cherry-red spot, muscle weakness, ataxia, intellectual disability, paralysis, early death	Altered lipid trafficking; impaired autophagy	[94,95,96,97]
Tay–Sachs Disease, AB Variant (AB-variant GM2)	*GM2A* GM2 ganglioside activator	GM2 ganglioside in neurons in the brain and spinal cord	Psychomotor deterioration, seizures, vision and hearing loss, intellectual disability, paralysis, cherry-red spot, early death	Impaired autophagy	[95,98,99]
Metachromatic leukodystrophy (MLD)	*ASA* or *PSAP* arylsulfatase A or prosaposin	Sulfatide compounds in neural tissue	Demyelination in central and peripheral nervous systems (peripheral neuropathy, mental retardation, motor dysfunction, ataxia, hyporeflexia), seizures, incontinence, paralysis, inability to speak, blindness, hearing loss	Affected trafficking due to altered chain length of the lipids; defective autophagosome–lysosome fusion, impaired autophagy	[100,101,102,103]
Sandhoff disease	*HEXB* β-hexosaminidase A and β-hexosaminidase B	GM2 ganglioside in neurons of the brain and spinal cord	Progressive nervous system deterioration, muscle weakness, ataxia, speech problems, mental retardation, blindness, seizures, spasticity, macrocephaly, cherry-red spots in the eyes, frequent respiratory infections, doll-like facial appearance, hepatosplenomegaly	Disruption of autophagy, aberrant lysosomal–autophagic turnover	[104,105,106,107]
Multiple sulfatase deficiency	*SUMF1* formylglycine-generating enzyme (FGE)	Sulfatides, sulfated glycosaminoglycans, sphingolipids and steroid sulfates in tissues	Leukodystrophy, movement problems, seizures, developmental delay, slow growth, ichthyosis, hypertrichosis, skeletal abnormalities (scoliosis, joint stiffness, dysostosis multiplex), hypotonia, coarse facial features, mild deafness, hepatomegaly, progressive neurologic deterioration, hydrocephalus	Accumulation of autophagosomes, defective autophagosome–lysosome fusion	[108,109,110]
GM1 gangliosidosis	*GLB1* β-galactosidase	GM1 ganglioside in tissues and organs, particularly in the brain	Hepatosplenomegaly, skeletal abnormalities, seizures, profound intellectual disability, cherry-red spot, gingival hypertrophy, cardiomyopathy, dysostosis multiplex, coarsened facial features	Accumulation of autophagosomes, impaired lysosomal flux	[101,111,112]
Schindler disease	*NAGA* α-N-acetylgalactosaminidase	Glycosphingolipids, glycoproteins and oligosaccharides with terminal or preterminal N-acetylgalactosaminyl residues in the lysosomes of most tissues	Developmental regression, blindness, seizures, loss of awareness of surroundings, unresponsive, cognitive impairment, sensorineural hearing loss, weakness and loss of sensation, angiokeratomas	No data	[113]
Sea-blue histiocytosis (inherited lipemic splenomegaly)	*APOE* apolipoprotein E	Cholesterol, triglycerides and beta-very-low-density lipoproteins (beta-VLDLs) in the blood; glycosphingolipids, particularly sphingomyelins in the histocytes	Hypertriglyceridemia, splenomegaly, liver function abnormalities, heart disease, sea-blue histiocytes in many organs (bone marrow, liver and spleen)	No data	[114]
*Neuronal ceroid lipofuscinosis*					
Batten disease (juvenile neuronal ceroid lipofuscinosis, CLN3 disease)	*CLN3* battenin, hydrophobic transmembrane protein involved in lysosomal function	Lysosomal autofluorescent storage material (AFSM) in the cells of the brain, central nervous system, and retina in the eye	Progressive blindness, seizures, mental and cognitive decline, dementia, speech and motor skills problems, premature death	Disruption of autophagy, vacuole maturation and impaired mitophagy; impaired autophagic clearance, defective autophagosome maturation	[115,116,117,118]
Jansky–Bielschowsky disease (late infantile neuronal ceroid lipofuscinosis, LINCL, CLN2 disease)	*TPP1* tripeptidyl-peptidase 1	Lipopigments in neurons, primarily in the cerebral and cerebellar cortices	Epilepsy, ataxia, myoclonus, vision loss, speech and motor skills problems (e.g., sitting and walking), developmental regression, intellectual disability, behavioral problems	Reduction in autophagic flux, inhibition of autophagosome formation, reduction in autophagosomes and autophagic degradation	[119,120]
*Lysosomal and lipase deficiency*					
Lysosomal acid lipase deficiency (Wolman disease, cholesteryl ester storage disease)	*LIPA* lysosomal acid lipase	Cholesteryl esters, triglycerides, and other lipids within lysosomes of most tissues	Hepatosplenomegaly, ascites, calcified adrenal glands, vomiting, diarrhea with steatorrhea, progressive psychomotor degradation, anemia, cachexia, low muscle tone, jaundice, vomiting, developmental delay, anemia, poor absorption of nutrients from food	Impairment of the lipophagic pathway	[121,122,123,124]
*Mucolipidosis*					
Mucolipidosis IV	*MCOLN1* (*TRPML1*) mucolipin-1	Sphingolipids, phospholipids, mucopolysaccharides and glycoproteins in cells of almost all tissues, including liver, spleen and in fibroblasts	Intellectual disability, psychomotor retardation, hypotonia, retinal degeneration, strabismus, photophobia, myopia, amblyopia or blindness, iron-deficiency anemia, achlorhydria with elevated blood gastrin levels	Impairment of autophagy and lipolysis; accumulation of lysosomes, autophagosomes and autophagy substrates	[125,126,127,128]
Sialidosis (mucolipidosis I)	*NEU1* neuraminidase 1	Sialic acid–containing compounds (sialyloligosaccharides and sialolipids) in lysosomes in bodily tissues	Type I: progressive neurological impairment without bone or joint abnormalities; type II: mental retardation, severe hepatosplenomegaly, coarse facial features, dysostosis multiplex, seizures, myoclonus, ataxia, aminoaciduria, corneal opacity, macular cherry-red spot, skeletal abnormalities	Impairment of lipolysis and autophagy	[129,130,131]
*Neutral lipid storage disease*					
Neutral lipid storage disease with myopathy	*PNPLA2* adipose triglyceride lipase (ATGL)	Triglycerides in muscle and other tissues	Myopathy, fatty liver, cardiomyopathy, pancreatitis, hypothyroidism, type 2 diabetes	Impairment of lipolysis	[132,133,134]
Chanarin–Dorfman syndrome (neutral lipid storage disease type I, neutral lipid storage disease with ichthyosis)	*ABHD5* abhydrolase domain containing 5 (activator of ATGL)	Triglycerides in organs and tissues, including skin, liver, muscles, intestine, eyes and ears	Ichthyosis, hepatomegaly, cataracts, ataxia, hearing loss, short stature, myopathy, nystagmus, mild intellectual disability	Impaired long-chain fatty acid oxidation; impaired BECN1-induced autophagic flux	[135,136,137]
*Xanthomatosis*					
Cerebrotendinous xanthomatosis (CTX)	*CYP27A1* sterol 27-hydroxylase	Cholestanol and bile alcohols in the blood	Neonatal cholestasis, childhood-onset cataract, tendon and brain xanthomata, neurologic dysfunction (dementia, psychiatric disturbances, pyramidal and/or cerebellar signs, seizures and neuropathy), liver dysfunction, intellectual impairment, neuropsychiatric symptoms (hallucinations, aggression and depression)	Induced autophagy	[138,139,140]
*Plant sterol storage disease*					
Sitosterolemia	*ABCG5* or *ABCG8* sterolin	Plant sterols, such as sitosterol, and LDL in the blood	Atherosclerosis, increased chance of a heart attack, stroke or sudden death, xanthomas, joint stiffness and pain, hemolytic anemia, macrothrombocytopenia	Accumulation of autophagic vacuoles	[141,142]
*Farber lipogranulomatosis*					
Farber disease (Farber lipogranulomatosis)	*ASAH1* acid ceramidase	Lipids in cells and tissues throughout the body, particularly around the joints.	Lipogranulomas, swollen and painful joint deformity, subcutaneous nodules, hoarseness, difficulty breathing, hepatosplenomegaly, developmental delay, vomiting	Impairment of autophagic flux	[143]
*Fucosidosis*					
Fucosidosis	*FUCA1* alpha-L-fucosidase	Fucose containing glyco-lipids and polysaccharides in the brain, liver, spleen, skin, heart, pancreas and kidneys	Intellectual disability, dementia, delayed development of motor skills, impaired growth, dysostosis multiplex, seizures, spasticity, angiokeratomas, coarse facial features, recurrent respiratory infections, visceromegaly	Induction of the autophagic cell death	[144]
**Lipid metabolism diseases**					
Familial hyperlipidemia					
*Hyperlipoproteinemia*					
Familial dysbetalipoproteinemia (hyperlipoproteinemia type III)	*APOE* apolipoprotein E	Chylomicrons and VLDL remnants in plasma	Palmar and tuberoeruptive xanthomas, coronary heart disease, peripheral vascular disease	Decreased lipolysis	[145,146,147,148]
Familial hypercholesterolemia (hyperlipoproteinemia type IIa)	*LDLR* LDL receptor	LDL in plasma	Tendon xanthomas, coronary heart disease, increased chance of a heart attack, stroke or sudden death	Impairment of autophagic flux; altered autophagy flux by persistent mitophagy	[149,150,151]
Familial defective apoB-100 (hyperlipoproteinemia type IIa)	*APOB* apolipoprotein B-100	LDL in plasma	Tendon xanthomas, coronary heart disease, increased chance of a heart attack, stroke or sudden death	Impairment of autophagic flux; altered autophagy flux by persistent mitophagy	[151,152]
*Familial chylomicronemia syndrome*					
ApoA-V deficiency	*APOA5* apolipoprotein A-V	Chylomicrons and VLDL in blood	Eruptive xanthomas, hepatosplenomegaly, pancreatitis	Impairment of lipolysis	[153,154]
GPIHBP1 deficiency	*GPIHBP1* glycosylphosphatidylinositol-anchored high-density lipoprotein binding protein 1	Chylomicrons in plasma	Eruptive xanthomas, pancreatitis	Impairment of lipolysis	[155,156]
Lipoprotein lipase deficiency (hyperlipoproteinemia type I)	*LPL* lipoprotein lipase	Chylomicrons in plasma	Eruptive xanthomas, abdominal pain, lipemia retinalis, hepatosplenomegaly, pancreatitis	Impairment of lipolysis	[157,158]
Familial apolipoprotein C-II deficiency (hyperlipoproteinemia type I)	*APOC2* apolipoprotein C-II (LPL cofactor)	Chylomicrons in plasma	Eruptive xanthomas, abdominal pain, lipemia retinalis, hepatosplenomegaly, pancreatitis	Impairment of lipolysis	[159,160,161]
Familial hepatic lipase deficiency	*LIPC* hepatic lipase	VLDL remnants and IDLs in plasma	Pancreatitis, coronary heart disease, increased chance of a heart attack, stroke or sudden death	Impairment of lipolysis	[162]
*Familial hypercholesterolemia*					
Autosomal recessive hypercholesterolemia	*LDLRAP1* (*ARH*) low-density lipoprotein receptor adaptor protein 1	LDL in plasma	Tendon xanthomas, coronary heart disease, increased chance of a heart attack, stroke or sudden death	Induced autophagy	[73,163,164]
Autosomal dominant hypercholesterolemia	*PCSK9* proprotein convertase subtilisin/kexin type 9	LDL in plasma	Tendon xanthomas, coronary heart disease, increased chance of a heart attack, stroke or sudden death	Increased autophagic flux	[73,165,166]

Groups (bold regular font), subgroups (regular font) or classes (italic font) of disorders related to abnormal lipid storage or lipid metabolism are indicated in the lines with a gray background. VLDL, Very Low Density Lipoprotein; LDL, Low Density Lipoprotei.

**Table 2 ijms-21-06113-t002:** Secondary lipid storage in lysosomal storage diseases. The individual classes of lipids are indicated in the lines with a gray background.

Secondary Storage Lipid	Disease	Compartment	Cellular Disturbance	Reference
Phospholipids				
Sphingomyelin	*Sphingolipidoses*: Niemann–Pick type C	Lysosomes	Altered membrane lipids trafficking	[170,171]
Bis(monoacylglycero)phosphate (BMP)	*Sphingolipidoses*: Niemann–Pick type C, Fabry disease, Gaucher disease, GM1 gangliosidosis, GM2 gangliosidosis *Mucopolysaccharidoses*: Hurler syndrome, Hunter syndrome *Neuronal ceroid lipofuscinoses*: NCL 10	Endosomes, lysosomes	Altered membrane lipids trafficking, lamellar bodies formation	[171,172]
Glycosphingolipids				
Gangliosides—GM1, GM2, GM3, GD1a, GD2, GD3	*Sphingolipidoses*: Niemann–Pick type A, B and C, Gaucher disease, prosaposin deficiency *Mucopolysaccharidoses*: Hurler syndrome, Hunter syndrome, Sanfilippo syndrome, Maroteaux–Lamy syndrome, Sly syndrome *Glycoproteinoses*: Galactosialidosis, α-mannosidosis, sialidosis *Mucolipidoses*: mucolipidosis II/III, mucolipidosis IV *Neuronal ceroid lipofuscinoses*: NCL 3, NCL 6, NCL 10	Late endosomes, lysosomes, cytoplasmic vesicles	Alteration of lysosomal pH, autophagy dysregulation, rupture of H^+^/Ca^2+^ homeostasis, altered vesicle trafficking, dysregulation of signaling pathways, accumulation of polyubiquitinated proteins, reduced capacity of immune cells to produce cytokines and antibodies, neurodegeneration (gliosis, demyelination of white matter, astrocyte and microglial activation)	[173,174,175,176,177,178,179,180,181]
Cholesterol				
Cholesterol	*Sphingolipidoses*: Niemann–Pick type A and B *Mucopolysaccharidoses*: Hurler syndrome, Hunter syndrome, Sanfilippo syndrome, Maroteaux–Lamy syndrome *Glycoproteinoses*: α-mannosidosis	Late endosomes, lysosomes, cytoplasmic vesicles	Impaired vesicle trafficking, abnormal sequestration of materials, foam cells in cerebral blood vessels and liver	[171,174,176,177,178]

Subgroups (regular font) or classes (italic font) of disorders related to abnormal lipid storage or lipid metabolism are indicated in the lines with a gray background.

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
