# Peer review of "Lipophagy and Lipolysis Status in Lipid Storage and Lipid Metabolism Diseases"

_ijms, 2020, doi:10.3390/ijms21176113_

Round 1
Reviewer 1 Report
Kloska et al provide a very comprehensive and thoughtful review article on lipophagy, lipolysis and lipid metabolism disease. This is a well-done review that is thorough and quite comprehensive and long, and it makes an important contribution to the field. However, there are some issues that I think need to be addressed prior to publication:
- While Figure 1 is very good, it would have been very helpful to have an additional figure on the transcriptional and functional regulation of lipophagy.
- The transition from the molecular and transcriptional regulation of lipophagy and lipolysis to the section on lysosomal storage diseases (LSDs) is quite disjointed. The authors should explain up front why these diseases are related to lipolysis and lipophagy. Those that are not related to these processes should be removed.
- There is a disconnect between LSDs and abnormal autophagy. Observing these two in concert does not directly implicate a role for lipophagy in the pathophysiology of these diseases, which is what is vaguely implied by discussing them in a review on lipophagy/lipolysis. The authors should explicitly state a role for lipolysis and/or lipophagy in each of these LSDs.
Minor:
- There are several grammatical errors throughout
- Section 4.3: The authors state that “Excessive accumulation of compounds in cellular compartments is always a pathological process.” This is too strongly worded and should not say “always.” I would recommend saying “often.”
Author Response
#Reviewer 1 Comments and Suggestions for Authors
Kloska et al provide a very comprehensive and thoughtful review article on lipophagy, lipolysis and lipid metabolism disease. This is a well-done review that is thorough and quite comprehensive and long, and it makes an important contribution to the field. However, there are some issues that I think need to be addressed prior to publication:
> While Figure 1 is very good, it would have been very helpful to have an additional figure on the transcriptional and functional regulation of lipophagy.
Answer: In the revised version of our manuscript, we have included three additional figures to better understand the problem:
Figure 2: “Energy release from saturated fatty acids in mitochondrial β-oxidation”
This figure is located on page 7 and is referred in the text in line 237.
Figure 3: “Transcriptional regulation of autophagy/lipophagy, lipolysis and lipid metabolism by transcription factor EB (TFEB) under nutrition-rich conditions (A) and nutrient depletion or aberrant lysosomal storage (B).”
This figure is located on page 8 and is referred in the text in lines 277 and 279.
Figure 4: “. Alterations in different stages of autophagy in the pathogenesis of lipid storage diseases”
This figure is located on page 21 and is referred in the text in line 436.
> The transition from the molecular and transcriptional regulation of lipophagy and lipolysis to the section on lysosomal storage diseases (LSDs) is quite disjointed. The authors should explain up front why these diseases are related to lipolysis and lipophagy. Those that are not related to these processes should be removed.
Answer: We have included a separate paragraph (4.2. Dysregulation of Autophagy or Lipolysis in Diseases) in section: “Lipid Metabolism and Diseases” (page 20, lines 428-445); this paragraph summarizes the role authophagy/lipophagy/lipolysis in the pathogenesis of diseases and we completed Table 1 and added a column “Perturbations in autophagy/lipophagy/lipolysis”. Additionally, we summarized the information on autophagy impairment in storage disorders in figure 4: “Alterations in different stages of autophagy in pathogenesis of lipid storage diseases” (page 21).
> There is a disconnect between LSDs and abnormal autophagy. Observing these two in concert does not directly implicate a role for lipophagy in the pathophysiology of these diseases, which is what is vaguely implied by discussing them in a review on lipophagy/lipolysis. The authors should explicitly state a role for lipolysis and/or lipophagy in each of these LSDs.
Answer: The information has been completed in the text (referred in the text in lines 428-445) and Table 1 and Figure 4.
Minor:
> There are several grammatical errors throughout
> Section 4.3: The authors state that “Excessive accumulation of compounds in cellular compartments is always a pathological process.” This is too strongly worded and should not say “always.” I would recommend saying “often.”
Answer: We have changed as suggested (line 492).
Reviewer 2 Report
Anna Kloska's paper discusses a very interesting and updated topic. The part of the paper concerning the molecular aspects is careful and clear.
While the section concerning the clinical aspects and the tables summarizing the pathologies dependent on lipid disorders are carent and partial.
The clinical section describes lipid accumulation disorders and lipid metabolism disorders (better classifiable as hyperlipidemias, with high blood lipid titers), while lipid myopathies (e.g. lipin 1, carnitine, ect) are completely overlooked.
It would be more correct to refer to a classification in relation to the tissues involved or to all the genes involved.
Alternatively, specify that you intend to treat only some disorders or refer to some organs or tissues. In this sense it could be useful to change the classification and / or the titles of the tables. It is necessary to review the references: e.g. Tang's Q.-Q. paper Lipid metabolism and diseases. Ski. Bull. 2016, 61, 1471-1472. is not available in Pub Med, maybe wrong?
The references seem incomplete, e.g. there are no papers by R.A. Coleman or Zechner, which are very important in this field.
Overall, the clinical section seems to be significantly reviewed.
Reviewer 3 Report
I liked the review in general, it is interesting and well documented. It is a subject that I am passionate about. During review the following concerns arose:
- The authors have forgotten to mention the important regulatory role of cholesterol in point 2.1. Fatty Acids and Cholesterol—Essential and Toxic. Cholesterol is required for cell cycle progression and mitosis completion (Curr Opin Pharmacol. 2012; 12(6):717-23. doi: 10.1016/j.coph.2012.07.001.)
- The authors have forgotten to mention Figure 1B in the text. However, they mention Figure 2B which does not exist (page 5 line 167).
- On page 5 where it named the enzyme HSL, it has been overlooked to mention that this enzyme is responsible for hydrolyzing also cholesteryl and retinyl esters. Moreover, in testis, HSL is the only esterase that can hydrolyze cholesteryl ester, and the loss of this activity results in cholesteryl ester and diacylglycerol accumulation.
- What is missing in this review is the explanatory diagrams and figures. Reading the review is interesting and well documented, but it is dense. So, diagrams and figures help the reader a lot in understanding what is described in the manuscript. As for example in the following points:
- 3.2. Energy Release from Fatty Acids
- 3.3. Transcriptional Regulation of Lipophagy, Lipolysis and Lipid Metabolism (Especially the TFEB-mTOR connection, downregulated gene expression)
- 4.4. mTOR–TFEB Signalizing Pathway and Dysregulation of Autophagy in Lipid Storage Diseases
Author Response
#Reviewer 3 Comments and Suggestions for Authors
I liked the review in general, it is interesting and well documented. It is a subject that I am passionate about. During review the following concerns arose:
> The authors have forgotten to mention the important regulatory role of cholesterol in point 2.1. Fatty Acids and Cholesterol—Essential and Toxic. Cholesterol is required for cell cycle progression and mitosis completion (Curr Opin Pharmacol. 2012; 12(6):717-23. doi: 10.1016/j.coph.2012.07.001.)
Answer: We have added information about regulatory role of cholesterol in cell cycle and mitosis with corresponding references (page 2, lines 68-72).
> The authors have forgotten to mention Figure 1B in the text. However, they mention Figure 2B which does not exist (page 5 line 167).
Answer: We have corrected the error.
> On page 5 where it named the enzyme HSL, it has been overlooked to mention that this enzyme is responsible for hydrolyzing also cholesteryl and retinyl esters. Moreover, in testis, HSL is the only esterase that can hydrolyze cholesteryl ester, and the loss of this activity results in cholesteryl ester and diacylglycerol accumulation.
Answer: We have completed the information with corresponding references (page 5, lines 172-176).
> What is missing in this review is the explanatory diagrams and figures. Reading the review is interesting and well documented, but it is dense. So, diagrams and figures help the reader a lot in understanding what is described in the manuscript. As for example in the following points:
- 3.2. Energy Release from Fatty Acids
- 3.3. Transcriptional Regulation of Lipophagy, Lipolysis and Lipid Metabolism (Especially the TFEB-mTOR connection, downregulated gene expression)
- 4.4. mTOR–TFEB Signalizing Pathway and Dysregulation of Autophagy in Lipid Storage Diseases
Answer: In the revised version of our manuscript, we have included three additional figures to better understand the problem:
Figure 2: “Energy release from saturated fatty acids in mitochondrial β-oxidation.”
This figure is located on page 7 and is referred in the text in line 237.
Figure 3: “Transcriptional regulation of autophagy/lipophagy, lipolysis and lipid metabolism by transcription factor EB (TFEB) under nutrition-rich conditions (A) and nutrient depletion or aberrant lysosomal storage (B).”
This figure is located on page 8 and is referred in the text in lines 277 and 279.
Figure 4: “Alterations in different stages of autophagy in the pathogenesis of lipid storage diseases”
This figure is located on page 21 and is referred in the text in line 439.
Round 2
Reviewer 2 Report
I have carefully read the authors' answer and take note of the changes made, in particular with regard to the tables, I agree with the publication in this form.